# MicroRNA-4516 in Urinary Exosomes as a Biomarker of Premature Ovarian Insufficiency

**DOI:** 10.3390/cells11182797

**Published:** 2022-09-07

**Authors:** Zobia Umair, Mi-Ock Baek, Jisue Song, Seona An, Seung Joo Chon, Mee-Sup Yoon

**Affiliations:** 1Department of Molecular Medicine, School of Medicine, Gachon University, Incheon 21999, Korea; 2Lee Gil Ya Cancer and Diabetes Institute, Gachon University, Incheon 21999, Korea; 3Gachon University Gil Medical Center, Department of Obstetrics and Gynecology, College of Medicine, Gachon University, Incheon 21565, Korea; 4Department of Health Sciences and Technology, GAIHST, Gachon University, Incheon 21999, Korea

**Keywords:** urinary exosome, micro-RNA, premature ovarian insufficiency, hsa-miR-4516, turner syndrome

## Abstract

Premature ovarian insufficiency (POI) is a typical disorder of amenorrhea that lasts for a minimum of four months in women < 40 years old and is typically characterized by reduced estrogen levels and elevated serum concentrations of follicle-stimulating hormone. We collected urine samples from two participant cohorts from Gil Hospital of Gachon University (Incheon, Korea): a sequencing cohort of 19 participants (seven patients with POI (POI patients without Turner syndrome), seven patients with Turner syndrome (POI patients with Turner syndrome), and five control individuals (age-matched controls with confirmed ovarian sufficiency)) and a validation cohort of 46 participants (15 patients with POI, 11 patients with Turner syndrome, and 20 control individuals). Among differentially expressed miRNAs, hsa-miR-4516 was significantly upregulated in patients with POI in both cohorts, independent of the presence of Turner syndrome. Moreover, the upregulation of miR-4516 was confirmed in the ovary—but not in the uterus—of a cyclophosphamide and busulfan-induced POI mouse model. This was accompanied by a decrease in STAT3 protein level, a predicted target of miR-4516, via miRTarBase2020. Our study provides compelling evidence that miR-4516 is highly expressed in patients with POI and POI mouse models, suggesting that miR-4516 is a diagnostic marker of POI.

## 1. Introduction

Premature ovarian insufficiency (POI) is a pathological condition of ovarian reserve exhaustion before the age of 40 years, manifesting as amenorrhea or oligomenorrhea, hypoestrogenism, and elevated serum follicle-stimulating hormone concentrations (>25 mIU/mL) [1]. The prevalence of POI is approximately 1%, with some variation depending on ethnicity [2]. Patients with POI have an increased risk of cardiovascular disease [3], osteoporosis [4], and cognitive impairment [5]. They may also have elevated total and cancer-specific mortality rates [6]. POI is a highly heterogeneous disease and originates from iatrogenic, karyotypic, and genetic factors. However, the etiology of POI remains unknown in approximately 90% of cases. The occurrence of POI in young women of reproductive potential is difficult to predict because clinical symptoms preceding the disease have not been reported. In patients with early ovarian insufficiency, attempts at pregnancy are made through re-transplantation and subsequent in vitro activation treatment after an ex vivo culture by harvesting whole ovaries or ovarian tissues/follicles. However, patients with POI whose ovarian function cannot be restored are unlikely to have successful results with these methods. Therefore, a diagnostic approach is sought to screen for the abnormal deterioration of ovarian function and advise patients with high risks of POI for family planning.

Turner syndrome is the most frequent hereditary cause of POI and is occasionally observed in women with POI, with symptoms typically appearing before menarche [7]. In karyotyping, Turner syndrome is characterized by an entire or partial deletion of one X-chromosome, resulting in oocyte loss throughout childhood [8]. Together with infertility, Turner syndrome is also associated with short height, delayed puberty, ovarian dysgenesis, hypergonadotropic hypogonadism, congenital cardiac abnormalities, endocrine disorders (such as type 1 and type 2 diabetes), osteoporosis, and immunological problems [9]. Most women with Turner syndrome have primary or secondary hypergonadotropic hypogonadism, which necessitates hormone replacement therapy [9]. Management of POI would benefit from a diagnostic marker that distinguishes POI induced by Turner syndrome from POI produced by other causes. 

Exosomes are functional vehicles that transport complex proteins, lipids, and nucleic acids between cells [10,11,12,13]. They are endosomal in origin, range in size from 30 to 200 nm, and are secreted from a given parental cell [14]. Exosomes can affect the behavior of nearby cells, their parental cell microenvironment, and the phenotype of distant cells and tissues, thereby causing systemic consequences [15]. Because exosomes may be found in most body fluids, analyzing exosomes in liquid samples collected by using non-invasive or minimally invasive procedures could detect pathological alterations that would otherwise necessitate a tissue biopsy [15]. Exosomes contain numerous microRNAs (miRNAs) [16]; miRNAs are single-stranded, noncoding 17–24-nucleotides-long RNAs that influence post-transcriptional gene silencing by binding to the 3′-untranslated regions (UTRs) or open reading frames of their target mRNAs [17]. These exosomes may include a disease-specific miRNA signature that can be a valuable diagnostic tool [16].

Besides their detection in ovaries, miRNAs derived from liquid biopsy samples, such as blood plasma and serum, have been evaluated for their function in POI [18,19]. Since the various homeostasis mechanisms may reflect changes in the blood and, consequently, changes in miRNA expressions, the urine may represent a better choice as a non-invasive liquid biopsy sample [20]. Moreover, urine samples have additional benefits of ease of handling, low cost, and availability in large volumes. However, miRNA expression profiling in urine as a possible biomarker for POI screening, early identification, or prognosis has not been investigated.

In this study, we identified urinary exosome miRNAs specific for POI, which can be used as potential diagnostic markers in patients with POI. We analyzed the miRNA expression profile of POI patients with or without Turner syndrome, isolated urinary exosomes, selected putative miRNAs, and verified them in a larger patient cohort and a POI mouse model. Considering that early detection of POI is vital for preserving fertility, diagnostic methods for screening patients with POI can improve the management of the condition. Herein, we have also identified miRNA biomarkers that can be potentially used to develop rapid and straightforward diagnostic methods for POI screening.

## 2. Materials and Methods

### 2.1. Sample Collection and Characterization 

All patients with POI/Turner syndrome and control individuals were women who visited the outpatient clinic in the Department of Gynecology of Gil Hospital, Gachon University (Incheon, Korea), from January 2019 to August 2021. Women younger than 40 years who did not have menstruation for at least six months with persistent hypergonadotropic hypogonadism in four-week intervals were diagnosed with POI. Women diagnosed with POI underwent serum karyotyping to exclude X-chromosome deletions and translocations, which can be inherited and cause POI symptoms in women after age 30. This karyotyping procedure was also performed to screen for Y chromosomes in women under 30. The patients underwent serum fragile X mental retardation 1 (FMR1) gene mutation studies. The duration of amenorrhea was defined as the period of no menstruation, calculated from the day of last menstruation until the first visit to the outpatient clinic. The control group included non-pregnant women with regular menstrual cycles and without abnormal gynecologic ultrasonography findings. None of the patients had a history of drug intake affecting glucose and lipid metabolism and no known medical conditions or diseases, such as polycystic ovarian syndrome, Cushing’s syndrome, congenital adrenal hyperplasia, androgen-secreting tumors, or endometriosis. The patients with POI were classified into two groups: those who were karyotypically confirmed as having Turner syndrome (Turner syndrome group) and those who did not (POI group). Blood samples were collected on the day of POI diagnosis before administration of any hormonal medication to the POI and Turner syndrome groups and the second to the fifth day of the menstrual cycle from the women in the control group. This study initially collected urine samples from seven patients with POI, seven patients with Turner syndrome, and five control individuals in the sequencing cohort. Additional samples were collected from 15 patients with POI, 11 patients with Turner syndrome, and 20 control individuals in the validation cohort. If the participants underwent their menstrual cycles, they were requested to collect morning urine samples on their non-menstrual cycle days.

### 2.2. Urine Processing and Exosome Isolation

Morning urine of the patients was processed within 4 h after voiding, as no degradation was observed within this period in pre-trials. Approximately 50–100 mL of urine was transferred to tubes and centrifuged for 10 min at 2000× *g* to remove any cells or cell debris. The supernatant was transferred to fresh tubes and stored at −80 °C or processed to isolate exosomes within a week. Exosomes from the urine were isolated by using an ExoQuick-TC Exosome Precipitation Solution Kit (EXOTC10A-1, System Biosciences, Berkeley, CA, USA). In brief, 250 μL ExoQuick solution was added to 1 mL of concentrated urine (1:4), mixed thoroughly, and incubated overnight at 4 °C. Then the mixture was centrifuged at 1500× *g* for 30 min at 4 °C, and the supernatant was removed. The exosome pellet was resuspended in 200 μL of PBS.

### 2.3. RNA Isolation

Total RNA was extracted from exosomes by using TRIzol reagent (15596026, Thermo Fisher Scientific, Waltham, MA, USA), according to the manufacturer’s instructions. RNA quality was assessed by using an Agilent 2100 bioanalyzer containing an RNA 6000 Pico Chip (Agilent Technologies, Amstelveen, The Netherlands), and RNA quantification was performed by using a Nanodrop 2000 Spectrophotometer system (Thermo Fisher Scientific).

### 2.4. MicroRNA Library Preparation and Sequencing

RNA sequencing was performed by E-Biogen, Inc. (Seoul, Korea). A microRNA library was constructed by using the NEB Next Multiplex Small RNA Library Prep Kit (E7330L, New England BioLabs, Inc., Beverly, MA, USA) according to the manufacturer’s instructions. Briefly, during library construction, 1 μg of total RNA from each sample was used to ligate the adaptors, and cDNA was synthesized by using reverse transcriptase with adaptor-specific primers. PCR was performed to ensure library amplification, and the libraries were cleaned by using a QIAquick PCR Purification Kit (Qiagen, Inc., Hilden, Germany) and AMPure XP beads (Beckman Coulter, Inc., Brea, CA, USA). The yield and size distribution of the small RNA libraries were assessed by using an Agilent 2100 Bioanalyzer for the high-sensitivity DNA assay (Agilent Technologies, Inc., Santa Clara, CA, USA). High-throughput sequences were produced by using a Next Seq500 system with single-end 75 sequencing (Illumina, San Diego, CA, USA).

### 2.5. Data Analysis

Sequence reads were mapped by using the Bowtie2 software tool to obtain a bam file (alignment file). Mature miRNA sequences were used as a reference for mapping. Read counts mapped on mature miRNA sequences were extracted from the alignment file by using Bedtools (v2.25.0, Salt Lake City, UT, USA) [21] and Bioconductor [22], which use the R statistical programming language (version 3.2.2; R Development Core Team, 2011, Vienna, Austria). Read counts were used to determine the miRNA expression levels. The quantile normalization method was used for comparisons between samples. In the miRNA target study, miRWalk 2.0 [23] was performed. Functional gene classification was performed by using DIANA-miRPath v3.0 (Lamia, Greece) [24].

### 2.6. Transmission Electron Microscopy

The isolated exosome samples were applied to copper grids coated solely with a thin carbon foil (01340, Ted Pella, Inc. Redding, CA, USA). After allowing the sample to absorb for 2 min and blotting off excess buffer solution onto Whatman filter paper, the samples were stained with 2% (*w*/*v*) uranyl acetate for 1 min. Distilled water was added for 1 min to remove the excess uranyl acetate, followed by drying for 15 min. The results were recorded by using a Bio-High Voltage EM system (JEM-1400 Plus at 120 kV and JEM-1000BEF at 1000 kV; JEOL Ltd., Tokyo, Japan) at the Korea Basic Science Institute in the Korea.

### 2.7. Exosome Particle Size Analysis

The size distribution of the isolated exosomes was evaluated by using a particle analyzer (Litesizer 500; Anton Paar, Graz, Austria). PBS was used to dilute the samples. The size of each sample was measured for 60 s in triplicate, and the captured data were analyzed by using Excel. 

### 2.8. Western Blot Analysis

Exosomes were lysed, using a lysis buffer (9803, Cell Signaling Technology, Danvers, MA, USA), for protein analysis. The supernatant was collected following microcentrifugation at 13,000× *g* for 10 min and boiled for 5 min in a sodium dodecyl sulfate sample buffer [25]. Proteins in the mouse ovaries were extracted by using a T-PER tissue protein extraction reagent supplemented with protease (05892791001, Roche, Mannheim, Germany) and phosphatase inhibitors (4906837001, Roche, Mannheim, Germany). Ovary tissues (30 mg) were homogenized in an extraction buffer, using steel beads and a TissueLyser II (Qiagen, Hilden, Germany) for 1–2 min at 30 Hz. The lysates were centrifuged at 13,000× *g* for 10 min at 4 °C to remove debris. The supernatant was mixed with a sodium dodecyl sulfate sample buffer and boiled for 5 min. Proteins were subjected to electrophoresis on sodium dodecyl sulfate–polyacrylamide gels and then transferred to polyvinylidene fluoride membranes (IPVH00010, Millipore, Billerica, MA, USA). The membranes were incubated with the primary antibodies following the manufacturer’s instructions. Horseradish peroxidase-conjugated secondary antibodies were detected by using an Immobilon Western Chemiluminescent HRP Substrate (WBKLS0500, Millipore, Billerica, MA, USA). The band intensity of the Western blot was quantified via densitometry, using ImageJ software. The following antibodies were used: anti-TSG101 (ab125011, 1:1000) and anti-tubulin (ab11304, 1:2000) from Abcam (Cambridge, MA, USA); anti-CD9 (sc-13118, 1:200), anti-CD63 (sc-5275, 1:200), anti-STAT3 (sc-8019, 1:500), and anti-p53 (sc-126, 1:500) from Santa Cruz Biotechnology (Dallas, TX, USA), and anti-Bax (2772, 1:1000) antibody from Cell Signaling Technology. Secondary antibodies (anti-mouse 115-035-003, anti-rabbit 211-002-171) were obtained from Jackson ImmunoResearch Laboratories Inc. (West Grove, PA, USA).

### 2.9. Quantitative Real-Time Polymerase Chain Reaction

Total RNA was extracted from the exosomes and ovary tissues by using TRIzol reagent (15596026, Thermo Fisher Scientific), according to the manufacturer’s instructions. The miRCURY LNA RT kit (339340, QIAGEN, Hilden, Germany) was used to synthesize cDNA from 5 ng of total RNA by using a 10× miRCURY RT Enzyme Mix with UniSp6 RNA spike-in (QIAGEN) in 20 μL reaction volumes. Quantitative real-time polymerase chain reaction (qRT-PCR) analysis was performed by using a CFX384 C1000 Thermal Cycler (Bio-Rad, Hercules, CA, USA) with a 2× miRCURY SYBER Green Master Mix (QIAGEN). RNA expression levels were normalized against UniSp6 [26]. The sequences of the primers used in this study are listed in Table 1. 

For real-time TaqMan analysis of hsa-miR-4516, a Taqman advanced miRNA cDNA synthesis kit (A28007, Applied Biosystems, Foster City, CA, USA) was used to synthesize cDNA for miRNAs by using a multiplex reverse transcription technique with TaqMan^®^ advanced microRNA assays (A25576, Applied Biosystems) for hsa-miR-4516 (478308) and ath-miR-159a (4427975).

### 2.10. Establishment of A POI Mouse Model

Eight-week-old female pathogen-free-grade C57B6L/N mice were obtained from Orient Bio Inc. (Seongnam, Korea). The mice were kept at a controlled temperature (20–22 °C) and lighting (12/12 h light/dark cycle) and were allowed free access to water and food. Experimental animals were randomly divided into two groups, namely control (*n* = 7) and cyclophosphamide (CTX)+ busulfan (BUS)-POI (*n* = 7), and three independent experiments were performed. To induce POI, the mice were intraperitoneally injected with 120 mg/kg CTX and 12 mg BUS/kg weekly for two weeks. The control group was injected with saline. CTX (C0768) and BUS (B2635) were purchased from Sigma-Aldrich (St. Louis, MO, USA). 

### 2.11. Hematoxylin-and-Eosin Staining and SIRIUS Red Staining of the Mouse Ovaries

The ovary tissues were fixed in 4% paraformaldehyde overnight, dehydrated, embedded in paraffin, and cut into 5 μm-thick sections. Every fifth section was stained with hematoxylin and eosin, as described previously [27]**,** and assessed total and atretic follicles from the entire area of a cross-sectioned ovary section from each mouse (Saline, *n* = 7; CTX-BUS, *n* = 7). Sirius red staining was performed to determine the presence of collagen fibers in the ovarian tissues. Images of hematoxylin-and-eosin and Sirius-red staining were randomly captured by using a Motic Easyscan Digital Slide Scanner (Motic Hong Kong Limited, Hong Kong, China).

### 2.12. TUNEL Assay and Ki67 Immunohistochemistry

Apoptosis in ovary tissues was detected by using a DeadEnd™ Fluorometric TUNEL System (G3250, Promega, Fitchburg, WI, USA), following the manufacturer’s instructions. The slides were examined by using an Olympus CKX3-Houn microscope (Olympus, Tokyo, Japan) after preparation with a mounting medium for fluorescence with DAPI (Vector Lab, Newark, CA, USA). The number of TUNEL-positive cells was counted from three random fields (500 μm × 500 μm) from each ovary cross-section and averaged for each mouse. For Ki67 immunohistochemistry, antigen retrieval was performed by incubating the sections in 0.01 M citrate buffer (pH 6.0) and applying intense microwave irradiation for 30 min following deparaffinization and rehydration. Endogenous peroxidase activity was quenched for 10 min with 3% H_2_O_2_, and nonspecific binding was blocked for 1 h with 10% normal goat serum. The sections were treated with anti-Ki67 primary antibodies (1:200; ab16667, Abcam) for 24 h at 4 °C. Antigens were visualized by using diaminobenzidine. Finally, the slides were dehydrated and mounted after counterstaining with hematoxylin. Images of Ki67 immunohistochemistry were randomly captured by using a Motic Easyscan Digital Slide Scanner (Motic Hong Kong Limited).

### 2.13. Statistical Analyses

The baseline clinical characteristics of the included patients were analyzed by using a one-way analysis of variance in IBM SPSS Statistics software, version 28.0. The mean ± standard deviation of at least three independent experiments was calculated (three to seven independent experiments for all figures). Dots represent the individual data points in all the quantitative data graphs. A two-tailed paired Student’s *t*-test in Excel or GraphPad Prism version 9.0 for Microsoft Windows (GraphPad Software, CA, USA) was used to evaluate the statistical significance of the data when necessary (GraphPad Software, CA, USA). The threshold for statistical significance was set at *p* < 0.05.

## 3. Results

### 3.1. Clinical Characteristics

We selected a sequencing cohort (7 patients with POI (POI patients without Turner syndrome), 7 patients with Turner syndrome (POI patients with Turner syndrome), and 5 control individuals (age-matched controls with confirmed ovarian sufficiency) and a validation cohort (15 patients with POI, 11 patients with Turner syndrome, and 20 control individuals). Table 1 summarizes the clinical features of all the 65 patients in this study. There were no statistically significant differences between groups in age, weight, body mass index, diastolic blood pressure, aspartate aminotransferase, alanine aminotransferase, total cholesterol (TC), triglyceride, low-density lipoprotein cholesterol, or prolactin levels. Patients with Turner syndrome were significantly shorter in height (*p* < 0.001) and had higher high-density lipoprotein cholesterol (*p* = 0.013) and follicle-stimulating hormone (*p* < 0.001) levels (Table 2). Only 2 out of 15 patients with POI patients in the validation cohort and all patients with Turner syndrome had primary amenorrhea (Table 2 and Table 3). Among the patients with secondary amenorrhea, the mean duration of amenorrhea was 37.8 months. A total of 3 of the POI patients of the validation cohort had chromosomal mosaicism, whereas the remaining 12 patients had a normal 46,XX karyotype. None of the patients with POI who did not have Turner syndrome had FMR1 gene mutations (Table 3 and Table 4).

### 3.2. Identification of Isolated Exosomes in the Urine

To characterize exosomes in human urine, we assessed their morphology and diameter, as well as the presence of exosome-enriched protein markers such as CD9, CD63, and TSG101. Transmission electron microscopy showed that urine exosomes exhibited round or cup-shaped morphologies, with diameters ranging from 95 to 160 nm (Figure 1A). According to nanoparticle tracking analysis, isolated exosomes showed a median diameter of 172.0515 nm, with the highest peak at 5–6% of the expected distribution profiles, as is consistent with pure exosome preparations (Figure 1B). No significant difference was observed in the frequency of exosomes between the groups (Figure 1B). The number of exosomes per unit of urine in the control was comparable to that in patients, as evidenced by the amount of total RNA (Figure 1C). The Western blot analysis revealed the presence of CD9, CD63, and TSG101 (Figure 1D).

### 3.3. Differential Expression of Urinary Exosome miRNAs in Patients with POI, Patients with Turner Syndrome, and Control Individuals

We performed miRNA deep sequencing by using urinary exosomes from patients with POI (*n* = 7), patients with Turner syndrome (*n* = 7), and the control individuals (*n* = 5) to examine the miRNA expression profiles of urinary exosomes in respective patients. We found 2588 miRNAs in human urinary exosomes from that sample set. The miRNA expression profiles were comparatively analyzed across the three groups (e.g., miRNAs expressed in the POI group versus the control group, the Turner syndrome group versus the control group, the Turner syndrome group versus the POI group, and both POI and Turner syndrome group versus the control group). We identified 33 differentially expressed miRNAs, with differential expression between any given two groups defined as a statistically significant difference in expression (*p* < 0.05) of at least 1.3 times, with a normalization value of 2 (Table 5).

The heatmap and scatter plots showed differentially expressed miRNAs in POI-, Turner-syndrome-, and control-group-specific clusters (Figure 2A–C). The heatmap showed that hsa-miR-4516 was the most highly upregulated miRNA in POI samples compared to the control individuals. Moreover, hsa-miR-4516 was significantly upregulated in patients with Turner syndrome with lower levels than in patients with POI. To understand the biological roles of the predicted target genes of the differentially expressed miRNAs, a Kyoto Encyclopedia of Genes and Genomes (KEGG) analysis was performed. This process uncovers the primary pathways in which the prospective target genes may be involved. Figure 2D shows the top 20 most significantly enriched pathways. The top five pathways are extracellular matrix–receptor interaction, Hippo signaling pathway, fatty acid biosynthesis, renal cell carcinoma, and proteoglycans in cancer. In addition, a gene category analysis of selected miRNAs demonstrated that most of the selected miRNAs are involved in uterine embryonic development, ovarian follicle development, and antral follicle development; all of these processes are closely related to the development of POI (Figure 2E).

### 3.4. Validation of the Differentially Expressed miRNAs via qRT-PCR

To further validate the miRNA profiling results of our RNA sequencing analysis, urinary exosomes isolated from the validation cohort (15 patients with POI, 11 patients with Turner syndrome, and 20 control individuals) were used to confirm the expression of selected miRNAs via miRNA sequencing. From the generated heatmap and significance plot, we selected 12 miRNAs that belonged to the following categories: Group 1, miRNAs that were highly differentially expressed in the POI or Turner syndrome group; and Group 2, substantial differences in expression in the POI or Turner syndrome group, but evenly expressed within the same patient group (e.g., the POI group or Turner syndrome group) (Appendix A). Quantitative RT-PCR results confirmed the substantial expression of 12 miRNAs in the validation cohort. Comparisons of the relative expression levels of selected miRNAs between the sequencing cohort and the validation cohort are shown in Table 6 and Table 7 and Appendix A. For 10 of the 12 miRNAs selected for the validation analysis, the expression patterns of patients with POI in the validation cohort were the same as the expression patterns of the POI group in the sequencing cohort: hsa-miR-16-5p, hsa-miR-29a-3p, hsa-miR-30b-5p, hsa-miR-99b-3p, hsa-miR-151a-5p, hsa-miR-200b-3p, hsa-miR-423-3p, hsa-miR-941, hsa-miR-4516, and hsa-miR-7847-3p (Table 6 and Appendix A). Among them, the expression rates of two miRNAs were significantly different between patients with POI and control individuals, as we had originally identified in the sequencing cohort: hsa-miR-4516 was significantly upregulated (*p* = 0.008), whereas hsa-miR-30b-5p was significantly downregulated (*p* = 0.026; Table 6 and Appendix A). Notably, hsa-miR-4516 was significantly upregulated (*p* = 0.022) in patients with Turner syndrome compared to control individuals to a lesser extent, similar to the RNA sequencing data (Table 7 and Appendix A). In patients with Turner syndrome, hsa-miR-20a-5p expression was substantially increased compared to the control group (*p* = 0.447), while hsa-miR-200b-3p expression was substantially decreased (*p* = 0.296), similar to the sequencing cohort results (Table 7 and Appendix A). Although the expression pattern was different compared to that of the sequencing cohort, the expressions of hsa-miR-29a-3p (*p* = 0.021) and hsa-miR-99-3p (*p* = 0.011) were significantly altered in validation cohort patients with Turner syndrome, compared to the control group (Table 7 and Appendix A). In conclusion, hsa-miR-4516 expression detected via qRT-PCR analysis of RNA from the validation cohort exactly mapped the RNA sequencing results of the sequencing cohort. Moreover, it was significantly upregulated in patients with POI and Turner syndrome compared to the control group (Figure 3A). The TaqMan PCR assay confirmed the upregulation of hsa-miR-4516 expression in patients with POI and Turner syndrome (Figure 3B).

### 3.5. Confirmation of miR-4516 Upregulation in the Ovaries of a Chemotherapy-Induced POI Mouse Mode

To evaluate miR-4516 as a biomarker of POI, we established a chemotherapy-induced POI mouse model by injecting the mice with CTX and BUS. There was no apparent death or toxicity in the POI mouse model, but the size of the ovaries was smaller than that in the saline-injected control mice (Appendix A). Compared to the saline-injected control mice, the POI group exhibited significantly reduced follicle numbers, increased ratio of atretic follicles in total follicles, and increased collagen deposition as proven by Sirius red staining (Figure 4A,B). Furthermore, TUNEL-positive cells were evident in the follicles of the POI groups (Figure 4C), whereas Ki67-positive cells were few (Figure 4D), implying that a POI mouse model was successfully established. Next, we assessed the expression of mmu-miR-4516 in the ovaries of the chemotherapy-induced POI mouse model; mmu-miR-4516 expression was significantly increased in the ovaries but not in the uteri in the POI mouse model compared to the control (Figure 4E). These results suggest that upregulation of mmu-miR-4516 expression is associated with pathological changes in the ovaries. To interpret the pathological relevance of mmu-miR-4516 overexpression, we analyzed the targets of miR-4516 by using miRTarBase2020, an experimentally validated miRNA-target interaction database [28]. STAT3 was identified as the target of miR-4516 with solid experimental evidence (Appendix A). The 3′-UTR of STAT3 contains two putative hsa-miR-4516 binding sites [29] (Appendix A). Consistent with this interaction prediction, STAT3 protein expression was significantly decreased in the ovaries of the chemotherapy-induced POI mouse model, together with the increase in the expression of apoptotic markers Bax and p53 (Figure 4F). Taken together, these results suggest that upregulating miR-4516 is closely related to POI pathology in tandem with a decrease in STAT3 expression.

## 4. Discussion

A longstanding demand exists for non-invasive testing methods for identifying POI biomarkers, which are expected to facilitate convenient screening, early diagnosis, and monitoring therapy responses. In this study, we investigated the clinical utility of urinary exosome-based miRNAs as non-invasive diagnostic and prognostic indicators in patients with POI and Turner syndrome. We first performed miRNA expression profiling in human urinary exosomes of patients with POI and Turner syndrome patients. We identified miR-4516 as a highly upregulated miRNA in the sequencing cohort, validation cohort, and chemotherapy-induced POI mouse model.

The involvement of miRNAs in the development and risk analysis of POI has been previously investigated by using serum and plasma. In contrast to previous studies, we analyzed human urinary exosomes. In this study, 2588 miRNAs were identified from the sequencing cohort, and 33 differentially expressed miRNAs were selected. Among the selected miRNAs, miR-151a-5p, miR-let-7c-5p, and miR-423-5p have been previously associated with POI. Expressions of miR-151a-5p and miR-let-7c-5p were reported in two independent selections of differentially expressed miRNAs in the plasma of patients with POI [18,19], although they changed in opposite trends. A polymorphism in miR-423 (miR-423C>A), a differentially expressed miRNA in our study, has been linked to a higher idiopathic risk of POI in a study involving a Korean cohort [30]. In addition, several previously reported miRNAs, such as miR-23a, miR-27a, miR-22-3p, miR-146a, miR-181a, miR-196a, miR-290-295, and miR-608 [31,32], were not among the selected differentially expressed miRNAs; miR-4516, the highly expressed miRNA in our selection, has not been previously reported in patients with POI or Turner syndrome. In this study, we confirmed the specificity of miR-4516 for POI by using an independent validation cohort and a well-established POI mouse model, thereby supporting miR-4516 as a potential diagnostic biomarker for POI. These discrepancies may have resulted from the different sampling techniques used in this study and the differences in patient grouping. Further investigation with a larger number of patient samples or another independent cohort could help confirm the specificity of miR-4516. In addition, future investigation is required to determine whether miR-4516 levels correlate with various disease-related ovarian dysfunctions such as type 1 diabetes mellitus (T1DM). T1DM shows hypogonadism, reduced ovary size, and decreased oocyte quality [33,34]. Although the mechanisms underlying these effects of diabetes on reproduction are not entirely understood, T1DM seems to impair granulosa cell–oocyte communication, mitochondrial dysfunction during meiosis, apoptosis of cumulus cells, and alteration of DNA methylation status within ovarian follicles [35,36,37,38]. The assessment of whether miR-4516 is a dominantly causative factor of POI would be necessary before using miR-4516 as a diagnostic marker of disease-related ovarian dysfunction.

We also observed inconsistencies between the sequencing and qRT-PCR results. The current diagnostic method of POI could not differentiate the severity or stage of POI, and the cause of POI is highly diverse, which can lead to discrepancies in exosomal miRNA expression between the cohorts. In addition, the sequencing cohort is small and limited compared to the validation cohort, and this may have caused inconsistencies between the cohorts. A future study with a larger cohort size is required for further confirmation.

To our knowledge, this is the first study to implicate hsa-miR-4516 as a highly upregulated miRNA in POI. In addition, miR-4516 has not been previously identified in urinary exosomes, according to a miRTarbase analysis. We predicted STAT3 as a target of miR-4516 in POI, using miRTarBase, an experimentally validated tool (Appendix A). There are two complementary regions between hsa-miR-4516 and the 3′-UTR of STAT3; the second miRNA-binding site at 1918–1925 bp of the STAT3 3′-UTR is substantially conserved across different species (Appendix A) [29]. The cytoplasmic protein STAT3 regulates gene transcription for survival, proliferation, and anti-apoptotic and immune response genes when activated in response to cytokines [39]. We observed that the upregulation of miR-4516 was accompanied by a decrease in STAT3 protein level and an increase in the expression of p53 and Bax in the chemotherapy-induced POI mouse model, thereby implying that miR-4516 might induce apoptosis by controlling STAT3 expression (Appendix A), which is supported by other reports [29,40]. In addition to the role of STAT3 in regulating apoptosis, STAT3 uniquely affects granulosa cells by controlling the coordinated activation of primordial follicles throughout reproductive life. Leukemia inhibitory factor stimulates ovarian follicles in vitro by activating STAT3 and then transcribing and translating SOCS4, which interacts with the cardiac troponin-like cytokine, poly(rC) binding protein 1, and malate dehydrogenase 1 [41]. However, STAT3 possibly contributes to the negative regulation of PTEN-Akt signaling, a critical mediator of the development of primordial and secondary follicles [42], thus leading to apoptosis [43] (Appendix A). However, we cannot exclude the possibility that decreased STAT3 expression resulted from the side effect of chemotherapy in the POI mice model. Therefore, the decrease in STAT3 protein level in POI transgenic mice models such as Fragile X premutation RNA-knockout mice must be confirmed by further investigation [44].

We identified miRNAs that could differentiate between patients with POI and Turner syndrome. We found that the expressions of hsa-miR-29a-3p and hsa-miR-30b-5p decreased, while that of hsa-miR-4516 increased, in patients with POI, whereas the expression of all three miRNAs increased in patients with Turner syndrome in the validation cohort (Figure 3 and Appendix A). An analysis of the expression patterns of these three miRNAs led us to distinguish patients with POI from those with Turner syndrome. In patients with POI without Turner syndrome, hsa-miR-4516 expression is increased, but the expression of hsa-miR-29a-3p and hsa-miR-30b-5p is decreased; in patients with Turner syndrome, the expression of hsa-miR-4516, hsa-miR-29a-3p, and hsa-miR-30b-5p are all increased. However, we could not examine this possibility with a Turner mouse model including X-monosomic mice (39, XO) that all have a single-sex chromosome because these mice are usually fertile, which is different from their human counterparts [45]. A further expanded human validation cohort is needed to develop a diagnostic method that uses a comparative analysis of the expression of these three miRNAs to screen for POI and Turner syndrome without having to rely on chromosomal examination.

Taken together, we propose that hsa-miR-4516 can be developed as a biomarker of POI within an easy and convenient diagnostic test using non-invasive sampling methods. The evaluation of the miR-4516 levels with the severity of an abnormal decline in ovarian functions is required for its use as a predictive marker for appropriate preparation in advance.

## Figures and Tables

**Figure 1 cells-11-02797-f001:**
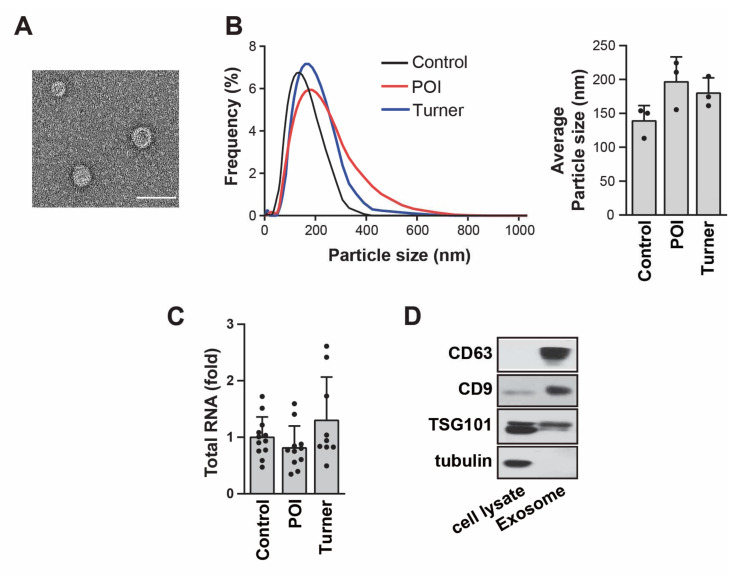
Characterization of exosomes isolated from the urine. (**A**) Phenotype of urinary exosomes detected via transmission electron microscopy. Scale bar: 100 nm. (**B**) Frequency of isolated urinary exosomes revealed via nanoparticle tracking analysis. All average particle sizes were calculated by using three independent samples. (**C**) The amount of total RNA from each sample was measured by using Nanodrop 2000 (ThermoFisher Scientific); patients with premature ovarian insufficiency (POI) = 11, patients with Turner syndrome = 9, control individuals = 12. (**D**) Expression levels of CD63, CD9, and TSG101 in urinary exosomes were detected via Western blot analysis. Tubulin was used as a negative control.

**Figure 2 cells-11-02797-f002:**
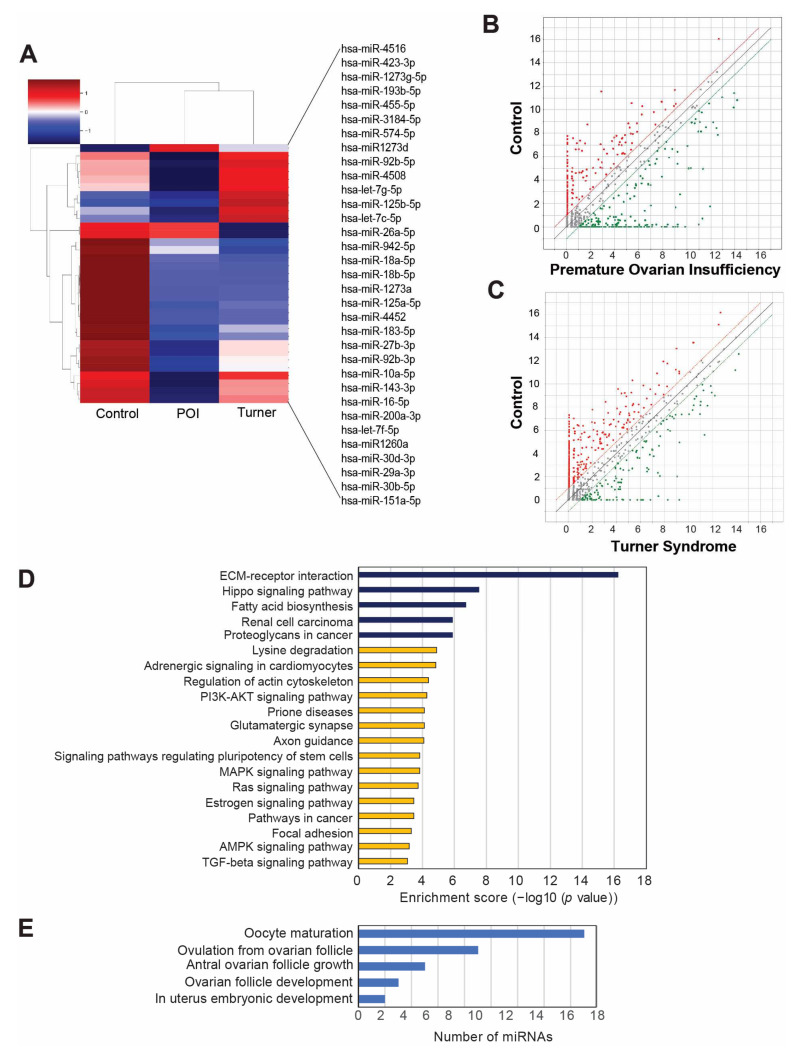
MicroRNA (miRNA) sequencing profile of urinary exosomes from patients with premature ovarian insufficiency (POI), Turner syndrome, and control individuals. (**A**) Heatmap with hierarchical clustering of 33 differentially expressed miRNAs (*p* < 0.05). Red and green colors represent increased or reduced expressions across samples, respectively. (**B**,**C**) Scatterplots of miRNA profiles among patients with POI versus the control (**B**) and patients with Turner syndrome versus the control (**C**). (**D**) Top 20 statistically significant enriched pathways in the Kyoto Encyclopedia of Genes and Genomes (KEGG) analyses of the genes targeted by the 33 differentially expressed miRNAs. (**E**) Gene category analysis for the POI-related target genes of the 33 differentially expressed miRNAs.

**Figure 3 cells-11-02797-f003:**
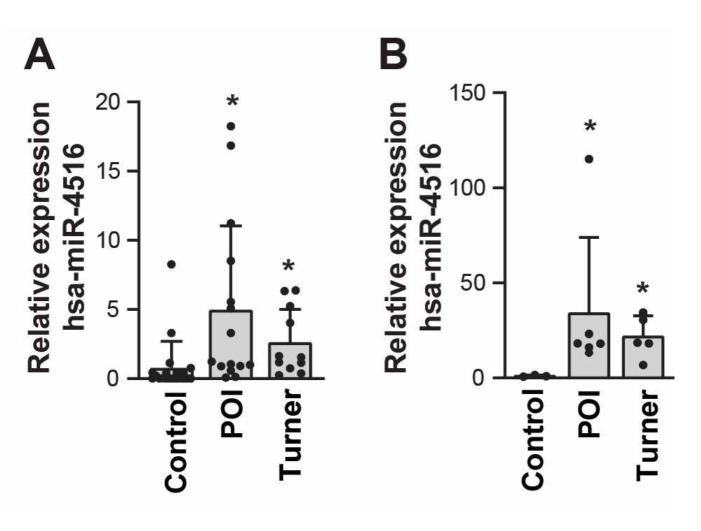
Relative expression of hsa-miR-4516 in the validation cohorts. (**A**) Relative expression of hsa-miR-4516 in the validation cohort, using qRT-PCR (patients with premature ovarian insufficiency (POI) = 15, patients with Turner syndrome = 11, and control individuals = 20). Relative miRNA expression was normalized by using UniSp6 as an internal control. (**B**) Relative expression of hsa-miR-4516, using real-time Taqman analysis, was normalized by ath-miR-159a as an endogenous control (patients with premature ovarian insufficiency (POI) = 6, patients with Turner syndrome = 5, and control individuals = 3). The *p*-value was evaluated by comparison to the control group. Asterisks (*) indicate significant differences (*p* < 0.05) when compared to the control group.

**Figure 4 cells-11-02797-f004:**
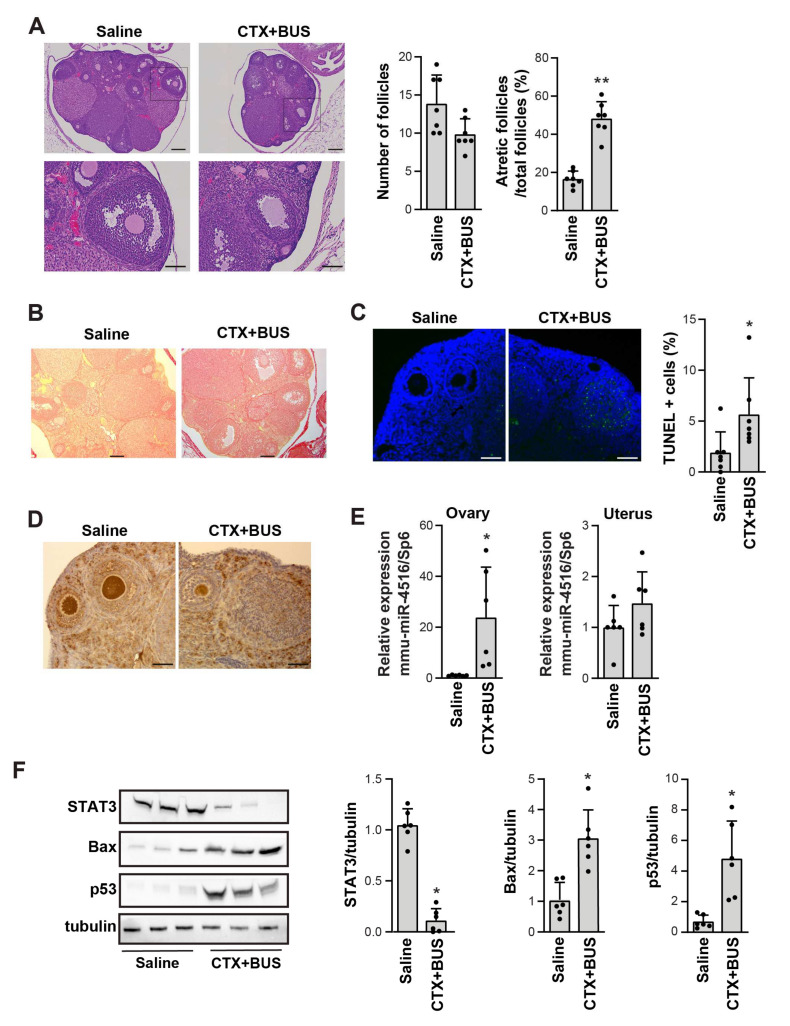
Increase in miR-4516 expression in a chemotherapy-induced premature ovarian insufficiency (POI) mouse model. (**A**) Representative hematoxylin-and-eosin-stained images of the ovaries from saline- and CTX+BUS-treated mice (*n* = 7). Scale bar = 200 μm in the upper panel and 80 μm in the lower panel. (**B**) Representative images of Sirius red staining in the ovaries from saline- and CTX+BUS-treated mice. Scale bar = 200 μm. (**C**) Representative images of the TUNEL assay in the ovaries from saline- and CTX+BUS-treated mice. Scale bar = 80 μm. TUNEL-positive cells were counted by using ImageJ software (*n* = 7). (**D**) Representative immunohistochemistry images of Ki67 expression in the ovaries from saline- and CTX+BUS-treated mice. Scale bar = 80 μm. (**E**) Relative expression of mmu-miR-4516 in the ovaries and uteri from the saline- and CTX+BUS-treated mice were analyzed by using qRT-PCR. Relative miRNA expression was normalized by using UniSp6 as an internal control (*n* = 6). (**F**) Western blot analysis of STAT3, Bax, and p53 expression in the ovaries of saline- and CTX+BUS-treated mice. The relative protein level in (**F**) was normalized to tubulin levels (mean ± SD; *n* = 6). Asterisks (*, **) indicate significant differences (*p* < 0.05, *p* < 0.01, respectively) when compared to saline-injected mice.

**Table 1 cells-11-02797-t001:** Sequences of the microRNA primers used in this study.

microRNA	Cat. No.	Sequence
hsa-miR-16-5p	YP00205702	UAGCAGCACGUAAAUAUUGGCG
hsa-miR-20a-5p	YP00204292	UAAAGUGCUUAUAGUGCAGGUAG
hsa-miR-29a-3p	YP00204698	UAGCACCAUCUGAAAUCGGUUA
hsa-miR-30b-5p	YP00204765	UGUAAACAUCCUACACUCAGCU
hsa-miR-99b-3p	YP00204064	CAAGCUCGUGUCUGUGGGUCCG
hsa-miR-151a-5p	YP00204007	UCGAGGAGCUCACAGUCUAGU
hsa-miR-200b-3p	YP00206071	UAAUACUGCCUGGUAAUGAUGA
hsa-miR-423-3p	YP00204488	AGCUCGGUCUGAGGCCCCUCAGU
hsa-miR-941	YP00204574	CACCCGGCUGUGUGCACAUGUGC
hsa-miR-4516	YP02112882	GGGAGAAGGGUCGGGGC
hsa-miR-4492	YP02100455	GGGGCUGGGCGCGCGCC
hsa-miR-7847-3p	YP02105945	CGUGGAGGACGAGGAGGAGGC

Abbreviations (Hsa, *Homo sapiens*; miR, microRNA).

**Table 2 cells-11-02797-t002:** Baseline clinical characteristics of the enrolled participants in the three groups.

	Sequencing Cohort				Validation Cohort			
	POI (*n* = 7)	Turner (*n* = 7)	Control (*n* = 5)	*p*-Value	POI (*n* = 15)	Turner (*n* = 11)	Control (*n* = 20)	*p*-Value
Age	37.7 ± 8.9	31.0 ± 6.1	34.8 ± 5.0	0.226	35.06 ± 9.92	34.00 ± 7.33	34.05 ± 17.11	0.052
Height (cm)	161.5 ± 6.2	148.1 ± 6.2	160.9 ± 7.3	0.002	162.39 ± 6.38	148.91 ± 5.27	163.81 ± 5.30	<0.001
Weight (kg)	56.9 ± 8.3	53.3 ± 6.4	54.8 ± 5.0	0.622	59.00 ± 2.46	55.41 ± 9.45	51.75 ± 10.41	0.197
BMI (kg/m^2^)	21.8 ± 2.7	24.4 ± 3.6	21.2 ± 1.5	0.105	22.48 ± 3.27	19.57 ± 4.27	19.24 ± 2.72	0.340
SBP (mmHg)	126.4 ± 3.4	135.1 ± 7.9	123.3 ± 3.8	0.003	124.31 ± 4.35	134.11 ± 11.12	122.00 ± 4.18	<0.001
DBP (mmHg)	75.1 ± 8.4	80.9 ± 7.3	80.0 ± 1.7	0.258	76.06 ± 6.17	79.89 ± 8.10	76.70 ± 4.04	0.217
AST (IU/L)	20.0 ± 7.7	27.1 ± 16.5	15.0 ± 2.9	0.162	21.00 ± 5.89	27.22 ± 14.32	20.85 ± 8.03	0.224
ALT (IU/L)	15.3 ± 6.3	37.7 ± 40.1	14.7 ± 4.3	0.165	16.44 ± 7.23	35.11 ± 35.59	16.50 ± 5.92	0.022
TC (mg/dL)	189.0 ± 39.7	202.1 ± 31.3	162.2 ± 19.0	0.101	185.81 ± 32.70	198.56 ± 20.38	185.70 ± 28.25	0.663
TG (mg/dL)	127.9 ± 25.2	222.9 ± 245.2	77.5 ± 22.5	0.219	161.56 ± 124.18	141.80 ± 47.27	110.30 ± 43.74	0.296
HDL-C (mg/dL)	73.9 ± 18.2	80.3 ± 17.5	56.8 ± 14.2	0.063	109.63 ± 41.12	105.04 ± 22.10	98.95 ± 17.56	0.013
LDL-C (mg/dL)	98.1 ± 31.6	91.9 ± 9.3	96.9 ± 11.7	0.839	109.63 ± 41.12	105.04 ± 22.10	98.95 ± 17.56	0.534
Prolactin (ng/mL)	8.5 ± 2.8	7.9 ± 2.2	9.1 ± 1.4	0.655	8.10 ± 2.86	7.76 ± 2.17	8.25 ± 2.47	0.854
TSH (mIU/mL)	1.2 ± 0.6	2.6 ± 1.1	1.9 ± 0.4	0.010	1.50 ± 1.07	2.86 ± 1.00	1.70 ± 1.25	<0.001
FSH (mIU/mL)	114.8 ± 35.1	52.8 ± 15.5	7.1 ± 0.7	<0.001	92.00 ± 32.96	50.46 ± 26.74	7.2 ± 1.72	<0.001
Estradiol (pg/mL)	24.6 ± 16.7	15.7 ± 4.5	39.6 ± 4.2	0.003	17.95 ± 12.58	19.72 ± 8.37	81.27 ± 5.27	<0.001

Abbreviations: POI, premature ovarian insufficiency; Turner, Turner syndrome; control, control individuals; BMI, body mass index; SBP, systolic blood pressure; DBP, diastolic blood pressure; AST, aspartate aminotransferase; ALT, alanine aminotransferase; TC, total cholesterol; TG, triglyceride; HDL-C, high-density lipoprotein cholesterol; LDL-C, low-density lipoprotein cholesterol; TSH, thyroid-stimulating hormone; FSH, follicle-stimulating hormone).

**Table 3 cells-11-02797-t003:** Menstrual history of the patients and serum chromosomal analysis of both premature ovarian insufficiency and Turner syndrome groups in the sequencing cohort.

	Menarche (Age)	Duration of Amenorrhea (Months)	Karyotype	FMR Gene Mutation
**POI**				
1	15	18	46,XX	Normal
2	13	24	47,XXX [39]/46,XX[1]	Normal
3	15	24	46,XX	Normal
4	16	36	46,XX	Normal
5	12	3	46,XX	Normal
6	15	5	46,XX	Normal
7	14	30	46,XX	Normal
**Turner Syndrome**				
1	no menarche	NA	46,X,i(X)(q10),22pstk+[16]/45,X,22pstk+[14]	Normal
2	no menarche	NA	46,X,i(X)(q10)	Normal
3	no menarche	NA	45,X	Normal
4	no menarche	NA	46,X,i(X)(q10)	Normal
5	no menarche	NA	46,X,i(X)(q10)[24]/45,X[6]	Normal
6	no menarche	NA	45,X	Normal
7	no menarche	NA	45,X	Normal

Abbreviations: FMR, fragile X mental retardation 1; NA, not available; POI, premature ovarian insufficiency.

**Table 4 cells-11-02797-t004:** Menstrual history of the patients and serum chromosomal analysis of both premature ovarian insufficiency and Turner syndrome groups in the validation cohort.

	Menarche (Age)	Duration of Amenorrhea (Months)	Karyotype	FMR Gene Mutation
POI				
1	no menarche	NA	46,XX	Normal
2	15	18	46,XX	Normal
3	1516	10	46,XX	Normal
4	1414	6	46,XX	Normal
5	1216	70	46,XX	Normal
6	1215	6	46,XX,16qh+	Normal
7	11	120	46,XX	Normal
8	13	24	47,XXX[39]/46,XX[1]	Normal
9	14	120	46,XX,1qh+	Normal
10	15	24	46,XX	Normal
11	13	12	46,XX	Normal
12	14	24	46,XX	Normal
13	12	36	46,XX	Normal
14	no menarche	NA	46,XX	Normal
15	14	22	46,XX	Normal
Turner Syndrome				
1	no menarche	NA	45,X	Normal
2	no menarche	NA	46,X,i(X)(q10)	Normal
3	no menarche	NA	46,X,i(X)(q10)[24]/45,X[6]	Normal
4	no menarche	NA	45,X	Normal
5	no menarche	NA	45,X[21]/46,X,idic(X)(q25)[9]	Normal
6	no menarche	NA	46,X,i(X)(q10)	Normal
7	no menarche	NA	45,X	Normal
8	no menarche	NA	45,X[17]/46,X,i(X)(q10)[13]	Normal
9	no menarche	NA	46,X,i(X)(q10),22pstk+[16]/45,X,22pstk+[14]	Normal
10	no menarche	NA	45,X/46,XX	Normal
11	no menarche	NA	46,X,idic(X)(q22.1)[28]/45,X[22]	Normal

Abbreviations: FMR, fragile X mental retardation 1; NA, not available; POI, premature ovarian insufficiency.

**Table 5 cells-11-02797-t005:** Differentially expressed microRNAs in the sequencing cohort.

ID	Gene Symbol	Fold Change				*p*-Value				Normalized Data (log2)			
		P/C	T/C	T/P	P_T/C	P/C	T/C	T/P	P_T/C	P/C	T/C	T/P	P_T/C
1	hsa-miR-4516	35.549	7.254	0.204	21.401	0.328	0.044	0.334	0.406	5.415	10.567	8.274	9.835
2	hsa-miR-423-3p	0.028	1.803	64.076	0.916	0.072	0.474	0.047	0.922	10.150	4.999	11.000	10.023
3	hsa-miR-1273g-5p	0.327	1.599	4.897	0.963	0.072	0.423	0.033	0.949	1.624	0.009	2.301	1.569
4	hsa-miR-193b-5p	0.080	2.208	27.536	1.144	0.262	0.379	0.034	0.890	7.935	4.295	9.078	8.129
5	hsa-miR-455-5p	0.074	2.640	35.441	1.357	0.253	0.282	0.027	0.761	3.757	0.010	5.158	4.198
6	hsa-miR-3184-5p	0.070	3.182	45.408	1.626	0.155	0.236	0.040	0.658	9.232	5.397	10.902	9.933
7	hsa-miR-574-5p	0.375	12.518	33.403	6.446	0.330	0.117	0.050	0.328	2.975	1.559	6.620	5.663
8	hsa-miR1273d	0.679	5.582	8.225	3.130	0.356	0.029	0.006	0.224	0.764	0.205	3.245	2.410
9	hsa-miR-92b-5p	0.185	6.274	33.901	3.229	0.158	0.150	0.049	0.424	2.667	0.233	5.317	4.359
10	hsa-miR-4508	0.361	5.372	14.860	2.867	0.422	0.148	0.048	0.419	5.041	3.573	7.466	6.560
11	hsa-let-7g-5p	0.837	0.206	0.246	0.521	0.656	0.005	0.032	0.121	10.435	10.178	8.157	9.496
12	hsa-miR-125b-5p	0.854	0.289	0.339	0.571	0.864	0.042	0.415	0.480	10.507	10.279	8.717	9.700
13	hsa-let-7c-5p	0.142	0.084	0.596	0.113	0.025	0.017	0.435	0.001	10.966	8.148	7.400	7.822
14	hsa-miR-26a-5p	0.403	0.245	0.609	0.324	0.081	0.026	0.284	0.007	11.854	10.542	9.826	10.228
15	hsa-miR-942-5p	0.251	0.227	0.905	0.239	0.162	0.150	0.307	0.038	2.140	0.144	0.000	0.074
16	hsa-miR-18a-5p	0.017	0.015	0.903	0.016	0.090	0.090	0.300	0.014	6.032	0.147	0.000	0.075
17	hsa-miR-18b-5p	0.151	0.150	0.994	0.151	0.079	0.079	0.267	0.011	2.734	0.009	0.000	0.004
18	hsa-miR-1273a	0.106	0.105	0.994	0.105	0.177	0.176	0.269	0.048	3.253	0.009	0.000	0.005
19	hsa-miR-125a-5p	0.105	0.109	1.032	0.107	0.100	0.095	0.974	0.017	12.196	8.946	8.992	8.969
20	hsa-miR-4452	0.199	0.203	1.020	0.201	0.102	0.104	0.463	0.018	2.337	0.008	0.037	0.023
21	hsa-miR-183-5p	0.110	0.136	1.234	0.123	0.116	0.125	0.854	0.027	6.792	3.606	3.909	3.766
22	hsa-miR-27b-3p	0.285	0.304	1.065	0.294	0.076	0.077	0.921	0.017	10.659	8.848	8.939	8.894
23	hsa-miR-92b-3p	0.006	0.008	1.274	0.007	0.083	0.083	0.537	0.012	9.062	1.672	2.021	1.857
24	hsa-miR-10a-5p	0.077	0.153	1.985	0.115	0.142	0.178	0.398	0.041	13.895	10.198	11.187	10.776
25	hsa-miR-143-3p	0.114	0.159	1.392	0.137	0.107	0.124	0.602	0.023	10.043	6.912	7.389	7.170
26	hsa-miR-16-5p	0.007	0.099	13.575	0.053	0.038	0.058	0.095	0.004	7.108	0.010	3.773	2.876
27	hsa-miR-200a-3p	0.025	0.165	6.657	0.095	0.101	0.164	0.230	0.029	10.343	5.012	7.747	6.949
28	hsa-let-7f-5p	0.348	0.554	1.593	0.451	0.026	0.256	0.447	0.056	10.729	9.206	9.878	9.581
29	hsa-miR1260a	0.046	0.162	3.540	0.104	0.108	0.156	0.073	0.028	5.331	0.882	2.706	2.065
30	hsa-miR-30d-3p	0.205	0.877	4.274	0.541	0.250	0.871	0.038	0.397	2.294	0.009	2.105	1.408
31	hsa-miR-29a-3p	0.001	0.141	248.695	0.071	0.136	0.197	0.093	0.043	10.994	0.205	8.163	7.169
32	hsa-miR-30b-5p	0.010	0.240	24.083	0.125	0.064	0.172	0.199	0.025	9.434	2.788	7.378	6.437
33	hsa-miR-151a-5p	0.105	0.524	4.970	0.315	0.019	0.248	0.045	0.027	4.103	0.857	3.170	2.434

Fold change is presented as miRNA expression in patients with POI compared to the control individuals. Abbreviations: P, premature ovarian insufficiency; T, Turner syndrome; C, control individuals; P_T, average expression level of premature ovarian insufficiency and Turner syndrome; Hsa, *Homo sapiens*; miR, microRNA.

**Table 6 cells-11-02797-t006:** Comparison of the relative expression patterns of selected microRNAs in patients with premature ovarian insufficiency.

miRNA	miRNA Sequencing			Validation Results Obtained Using qRT-PCR		
	Up/Down	Fold Change	*p*-Value	Up/Down	Fold Change	*p*-Value
hsa-miR-16-5p	Down	0.007	0.038	Down	0.203	0.170
hsa-miR-20a-5p	Down	0.009	0.101	Up	2.410	0.361
hsa-miR-29a-3p	Down	0.001	0.136	Down	0.186	0.066
hsa-miR-30b-5p	Down	0.010	0.064	Down	0.218	0.026
hsa-miR-99b-3p	Down	0.060	0.254	Down	0.505	0.185
hsa-miR-151a-5p	Down	0.105	0.019	Down	0.813	0.751
hsa-miR-200b-3p	Down	0.002	0.096	Down	0.530	0.439
hsa-miR-423-3p	Down	0.028	0.072	Down	0.089	0.153
hsa-miR-941	Down	0.064	0.254	Down	0.283	0.065
hsa-miR-4492	Up	23.091	0.344	Down	0.276	0.094
hsa-miR-4516	Up	35.549	0.328	Up	4.971	0.008
hsa-miR-7847-3p	Up	10.175	0.085	Up	1.449	0.568

These patterns were obtained by using miRNA sequencing and validated by using qRT-PCR; Fold change is presented as miRNA expression in patients with POI compared to the control individuals. Abbreviations: Hsa, *Homo sapiens*; miR, microRNA; qRT-PCR, quantitative real-time polymerase chain reaction.

**Table 7 cells-11-02797-t007:** Comparison of the relative expression patterns of selected microRNAs in patients with Turner syndrome.

miRNA	miRNA Sequencing			Validation Results by qRT-PCR		
	Up/Down	Fold Change	*p*-Value	Up/Down	Fold Change	*p*-Value
hsa-miR-16-5p	Down	0.099	0.058	Up	1.673	0.811
hsa-miR-20a-5p	Up	1.014	0.987	Up	1.843	0.447
hsa-miR-29a-3p	Down	0.141	0.197	Up	9.788	0.021
hsa-miR-30b-5p	Down	0.240	0.284	Up	5.380	0.119
hsa-miR-99b-3p	Up	14.081	0.171	Down	0.175	0.011
hsa-miR-151a-5p	Down	0.524	0.248	Up	1.008	0.991
hsa-miR-200b-3p	Down	0.692	0.671	Down	0.439	0.296
hsa-miR-423-3p	Up	1.803	0.474	Down	0.345	0.363
hsa-miR-941	Up	5.138	0.235	Down	0.207	0.064
hsa-miR-4492	Up	13.250	0.058	Down	0.286	0.107
hsa-miR-4516	Up	7.254	0.044	Up	2.663	0.022
hsa-miR-7847-3p	Up	12.252	0.212	Down	0.286	0.026

These patterns were obtained by using miRNA sequencing and validated by using qRT-PCR; Fold change is presented as miRNA expression in patients with Turner syndrome compared to the control individuals. Abbreviations: Hsa, *Homo sapiens*; miR, microRNA; qRT-PCR, quantitative real-time polymerase chain reaction.

## Data Availability

The datasets GENERATED for this study can be found in the Gene Expression Omnibus, https://www.ncbi.nlm.nih.gov/geo/query/acc.cgi?acc=GSE205700 (accessed on 27 July 2022).

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
