# Peer review of "MicroRNA-4516 in Urinary Exosomes as a Biomarker of Premature Ovarian Insufficiency"

_cells, 2022, doi:10.3390/cells11182797_

Round 1

Reviewer 1 Report

This manuscript titled by “MicroRNA-4516 in urinary exosomes as a biomarker of premature ovarian insufficiency.” present urinary exosome miRNAs specific for POI, which can be used as potential diagnostic markers in patients with POI. The authors analyzed exosomal miRNAs from urine samples of POI and Turner syndrome patients. The authors initially used RNA-sequencing to find the miRNAs that differed in expression between POI, Turner patients, and healthy controls. They next used qPCR to confirm the expression of these miRNAs in a broader validation cohort and a POI mice model. The authors demonstrated the potential of miR-4516 as a non-invasive diagnostic marker using human and animal models. I think this study is a relatively well-organized and authors performed their study very well. A range of experiments are utilized to validate the biological significance of of miR-4516 and the results presented in the manuscript justified the conclusions drawn. I would strongly recommend this manuscript for publication after the minor corrections.     

 1. Authors collected morning urine to isolate exosomes in this study. Are there any reasons to use morning urine instead of 24-hour urine?

 2. The authors need to confirm miR-4516 expression in the patients with POI and Turner syndrome in the Taqman method.

 3. In Discussion (line 493-495), the authors suggested the expression pattern of hsa-miR-29a-3p, hsa-miR-30b-5p, and hsa-miR-4516 could be a diagnostic marker of the presence of Turner syndrome in POI patient. The graph to show the expression of hsa-miR-29a-3p and hsa-miR-30b-5p in the validation cohort with all the patients, such as miR-4516 in Figure 3, is required to support their suggestion.

4. 'hsa-miR-4516' in Figure 4E needs to be corrected as ‘mmu-miR-4516’.

Author Response

1. Authors collected morning urine to isolate exosomes in this study. Are there any reasons to use morning urine instead of 24-hour urine?

The best urine sample for isolating exosomes is typically the first morning void, according to general consensus1,2. Due to prolonged bladder retention, morning urine would typically be a more practical alternative because it is concentrated. In addition, morning urine correlates well with a 24-hour urine collection to measure urinary albumin excretion. It is less affected by factors such as hydration status and physical activity, reducing the variability caused by these factors3. Collecting 24-hour urine samples in an out-patient clinic could result in greater bias, which would also affect quality.

2. The authors need to confirm miR-4516 expression in the patients with POI and Turner syndrome in the Taqman method.

As per your suggestion, we performed qRT-PCR using the Taqman method. Taqman analysis confirmed the high expression of miR-4516 in patients with POI and Turner syndrome (Figure 3B).

3. In Discussion (line 493-495), the authors suggested the expression pattern of hsa-miR-29a-3p, hsa-miR-30b-5p, and hsa-miR-4516 could be a diagnostic marker of the presence of Turner syndrome in POI patient. The graph to show the expression of hsa-miR-29a-3p and hsa-miR-30b-5p in the validation cohort with all the patients, such as miR-4516 in Figure 3, is required to support their suggestion.

As per your suggestion, we added the graph depicting the expression of hsa-miR-29a-3p and hsa-miR-30b-5p in the validation cohort with all data points from all individuals (Figure S4).

4. 'hsa-miR-4516' in Figure 4E needs to be corrected as ‘mmu-miR-4516’.

We corrected it in Figure 4E.

All changes to the manuscript are in red.

We also uploaded the attachment.

Reviewer 2 Report

The authors using urine samples from patients with POI and Turner have conducted study to examine exosomal miR as potential early biomarker for POI. The authors conclude miR4516 could be developed into non-invasive biomarker and provide corroborative evidence using chemo-induced POI mouse model. The study appears to be pilot in nature but results are encouraging.

The following are my minor concerns and some questions for authors to address.

Did exosome enriched urine fraction remain in -80C for similar duration and does freeze thaw affect the quality of exosomes and if yes then how was it controlled in this study. 

Exosome size distribution plot is interesting but it is not clear if there were differences in size distribution between the groups. It is not clear what the frequency of exosomes per unit urine in these cohorts. 

The amount of RNA obtained per unit of urine should be presented to understand if possible. This will help understand quantitative differences.

There seems to be inconsistency between Table-6 and Figure-3 average values of miR4516.

Authors need to further discuss inconsistencies between p values in mir sequencing vs qRTPCR results in POI and Turner. The validation of markers showing significant opposite trends is concerning. Is it possible to validate miR4516 using another independent cohort? 

The limitations of the study needs to be further elaborated in the discussion.

Author Response

1. Did exosome enriched urine fraction remain in -80C for similar duration and does freeze thaw affect the quality of exosomes and if yes then how was it controlled in this study. 

We had considered the importance of the time duration of storing urine samples at -80°C. Accordingly, we isolated the exosomes within a week of collecting the individuals' urine. We have added this information in “Materials and Methods” on line 127.

2. Exosome size distribution plot is interesting but it is not clear if there were differences in size distribution between the groups. It is not clear what the frequency of exosomes per unit urine in these cohorts. 

We found no significant differences in the frequency of exosomes per unit urine between groups (Figure 1B). We added this description to “result” on lines 326-327.

3. The amount of RNA obtained per unit of urine should be presented to understand if possible. This will help understand quantitative differences.

Total RNA from the isolated exosomes was quantified as shown in Figure 1C. The amount of total RNA in exosomes from patients with POI or Turner syndrome was comparable to the control group. This result showed no significant difference in the number of exosomes between the groups (line 328-329).

4. There seems to be inconsistency between Table-6 and Figure-3 average values of miR4516.

We corrected the average value of miR-4516 in Figure 3 to match Table 6.

5. Authors need to further discuss inconsistencies between p values in mir sequencing vs qRTPCR results in POI and Turner. The validation of markers showing significant opposite trends is concerning. Is it possible to validate miR4516 using another independent cohort? The limitations of the study need to be further elaborated in the discussion. 

The variation of the sequencing and the validation cohort may have caused inconsistencies between the sequencing and qRT-PCR results. The current diagnosis method of POI could not differentiate the severity or stage of POI, and the cause of POI is highly diverse, which can lead to discrepancies in the exosomal miRNA expression between the cohorts. In addition, the sequencing cohort is small and limited compared to the validation cohort, which may have caused inconsistencies between the cohorts. A future study with a larger cohort size is required for further confirmation.

Notably, the current validation cohort was independent of the sequencing cohort. The specificity of miR-4516 for POI was confirmed using an independent validation cohort and a well-established POI mouse model, thereby supporting miR-4516 as a potential diagnostic biomarker for POI. As you suggested, it would be more conclusive if we validated miR-4516 in another independent cohort. However, collecting human samples in outpatient clinics for another independent cohort was not feasible to meet the revision deadline. Further investigation with a larger number of patient samples or another independent cohort could help confirm the specificity of miR-4516.

We added this to “Discussion” on lines 533-539 and 549-555 as follows.

In this study, we confirmed the specificity of miR-4516 for POI using an independent validation cohort and a well-established POI mouse model, thereby supporting miR-4516 as a potential diagnostic biomarker for POI. These discrepancies may have resulted from the different sampling techniques used in this study and the differences in patient grouping. Further investigation with a larger number of patient samples or another independent cohort could help confirm the specificity of miR-4516.

We also observed inconsistencies between the sequencing and qRT-PCR results. The current diagnostic method of POI could not differentiate the severity or stage of POI, and the cause of POI is highly diverse, which can lead to discrepancies in exosomal miRNA expression between the cohorts. In addition, the sequencing cohort is small and limited compared to the validation cohort, which may have caused inconsistencies between the cohorts. A future study with a larger cohort size is required for further confirmation.

1          Chen, C. Y., Hogan, M. C. & Ward, C. J. Purification of exosome-like vesicles from urine. Methods Enzymol 524, 225-241 (2013). https://doi.org:10.1016/B978-0-12-397945-2.00013-5

2          Gonzales, P. A. et al. Isolation and purification of exosomes in urine. Methods Mol Biol 641, 89-99 (2010). https://doi.org:10.1007/978-1-60761-711-2_6

3          Witte, E. C. et al. First morning voids are more reliable than spot urine samples to assess microalbuminuria. J Am Soc Nephrol 20, 436-443 (2009). https://doi.org:10.1681/ASN.2008030292

All changes to the manuscript are in red.

Round 2

Reviewer 2 Report

Authors have adequately addressed my concerns. No further comment.